# Cas9-mediated endogenous plasmid loss in *Borrelia burgdorferi*

**Constantin N. Takacs**[1,2,3], **Yuko Nakajima**[4], **James E. Haber**[4], **Christine Jacobs-Wagner**[1,2,3]*

**1** Department of Biology, Stanford University, Palo Alto, California, United States of America, **2** Sarafan ChEM-H Institute, Stanford University, Palo Alto, California, United States of America, **3** Howard Hughes Medical Institute, Stanford University, Palo Alto, California, United States of America, **4** Department of Biology and Rosenstiel Basic Medical Sciences Research Center, Brandeis University, Waltham, Massachusetts, United States of America

* jacobs-wagner@stanford.edu

**Data Availability Statement:** The DNA Sequence for strain K2's vls locus was deposited with NCBI GenBank (accession number OP620651).

**Funding:** C.N.T. was supported in part by an American Heart Association postdoctoral

## Abstract

The spirochete *Borrelia burgdorferi*, which causes Lyme disease, has the most segmented genome among known bacteria. In addition to a linear chromosome, the *B. burgdorferi* genome contains over 20 linear and circular endogenous plasmids. While many of these plasmids are dispensable under in vitro culture conditions, they are maintained during the natural life cycle of the pathogen. Plasmid-encoded functions are required for colonization of the tick vector, transmission to the vertebrate host, and evasion of host immune defenses. Different *Borrelia* strains can vary substantially in the type of plasmids they carry. The gene composition within the same type of plasmid can also differ from strain to strain, impeding the inference of plasmid function from one strain to another. To facilitate the investigation of the role of specific *B. burgdorferi* plasmids, we developed a Cas9-based approach that targets a plasmid for removal. As a proof-of-principle, we showed that targeting wild-type Cas9 to several loci on the endogenous plasmids lp25 or lp28-1 of the *B. burgdorferi* type strain B31 results in sgRNA-specific plasmid loss even when homologous sequences (i.e., potential sequence donors for DNA recombination) are present nearby. Cas9 nickase versions, Cas9$^{D10A}$ or Cas9$^{H840A}$, also cause plasmid loss, though not as robustly. Thus, sgRNA-directed Cas9 DNA cleavage provides a highly efficient way to eliminate *B. burgdorferi* endogenous plasmids that are non-essential in axenic culture.

## Introduction

Lyme disease, also known as Lyme borreliosis, is the most prevalent vector-borne disease in North America and Eurasia [1, 2]. It is caused primarily by the spirochete *Borrelia burgdorferi* and the related *Borrelia afzelii* and *Borrelia garinii* species. The disease presents with various symptoms that can include fever, malaise, rash, arthritis, neurological dysfunctions, and cardiac manifestations [3]. Humans are dead-end hosts. In nature, *B. burgdorferi* is maintained through a transmission cycle between a vertebrate host reservoir (e.g., white footed mice and other small mammals, but also birds) and an ixodid tick vector [4]. During feeding, *B.*

fellowship (award number 18POST33990330). C.
J.-W. is a Howard Hughes Medical Institute
Investigator. Y.N. was supported by the Bay Area
Lyme Foundation and the Brandeis University
Provost's Research Fund. J.E.H. was supported by
NIH grant R35 GM127029. The funders had no role
in study design, data collection, analysis, and
interpretation, decision to submit the work for
publication, or preparation of the manuscript.

**Competing interests:** The authors have declared
that no competing interests exist.

*burgdorferi*-colonized tick vectors deliver the spirochetes into vertebrate hosts, where the spirochetes can replicate, disseminate, and often establish persistent infection.

Members of the Borreliaceae family contain the most segmented bacterial genomes known to date [5]. For instance, the genome of the *B. burgdorferi* type strain B31 is composed of a linear chromosome and 21 linear and circular plasmids [6, 7]. During growth, the Borreliaceae are polyploid, as each cell carries multiple copies of both the chromosome and plasmids [8, 9]. The chromosome encodes the vast majority of essential housekeeping and metabolic functions [6, 10]. In contrast, the plasmids primarily encode lipoproteins that mediate the spirochetes' interaction with the vertebrate and tick host environments and help them evade host immune defenses [6, 7, 11–16]. Additionally, each strain hosts several highly similar plasmid members of the cp32 class, which are prophages [7, 17–20]. In the *B. burgdorferi* type strain B31, which is the most well studied genetically, only plasmid cp26 has been shown to be required for growth in axenic culture [21–24]. Several other plasmids are known to be required in the vertebrate or tick hosts [4, 25, 26]. However, much remains unknown about the roles of *B. burgdorferi* plasmids. Furthermore, as the number of distinct plasmid types and the genes carried by any given plasmid type vary significantly among Borreliaceae species and strains [10, 17, 27, 28], strain-to-strain inferences of plasmid function are not always possible.

An effective way to investigate plasmid function is to remove it from a given strain. Spontaneous plasmid loss during extended passaging in axenic culture has been known since the early days of Lyme disease research [29, 30], but this approach is not specific to a particular plasmid of interest and often results in loss of multiple plasmids [25, 26, 31, 32]. Curing a specific plasmid can be achieved through transformation of *B. burgdorferi* with a shuttle vector that carries the plasmid maintenance locus of the endogenous plasmid of interest [33]. The incompatibility that arises between the endogenous plasmid and the introduced shuttle vector leads to displacement of the endogenous plasmid by the shuttle vector [33–39]. However, this approach requires knowledge of the plasmid maintenance locus of the targeted endogenous plasmids.

A more streamlined method to eliminate endogenous plasmids from *B. burgdorferi* strains would be to generate site-specific DNA lesions. In the absence of efficient DNA repair, those lesions might lead to degradation of the targeted endogenous plasmid. Indeed, in the absence of a recombinational donor sequence, exogenously induced double-stranded DNA breaks (DSBs) in the chromosome can be lethal in several bacteria, including *Escherichia coli* [40, 41], streptococci [42], *Clostridium cellulolyticum* [43], and the spirochete *Leptospira biflexa* [44]. However, DSBs can be repaired in some species, such as mycobacteria, by nonhomologous end-joining [45]. Repair of a site-specific DSB in *Neisseria gonorrhoeae*, when there are no homologous sequences to provide a template for recombinational repair, occurs at such low frequencies that less than one cell in ten thousands survives this type of genome lesion [46]. In contrast, the presence of short (5 to 23 base pairs) homologous sequences flanking an endonuclease-induced DSB led to RecA-mediated repair in a small fraction of cells [46]. Since most *B. burgdorferi* plasmids are not needed for growth in axenic culture, induction of DNA lesions in *B. burgdorferi* plasmids should cause plasmid loss if DNA repair is inefficient.

To generate such site-specific lesions, we used the clustered regularly interspaced palindromic repeats (CRISPR)-Cas9 system derived from *Streptococcus pyogenes* [47, 48]. Cas9 is the endonuclease component of a type of bacterial immune defense against invading foreign DNA molecules [49]. It has two catalytic residues, D10 and H840, each cutting one of the strands of the targeted double stranded DNA sequence [47]. Cas9 targeting to a specific DNA sequence can be achieved by co-expression of a short guide RNA molecule, or sgRNA. Base pairing between the Cas9-bound sgRNA and the target DNA sequence next to a protospacer-adjacent motif (PAM) directs the Cas9 activity to the genome location specified by the sgRNA

[47]. While wild-type Cas9 (Cas9$^{WT}$) generates a DSB in the target DNA sequence, single active site mutants (Cas9$^{D10A}$ and Cas9$^{H840A}$) are nickases that generate single-stranded DNA breaks (SSBs) [47]. Finally, the double mutant, catalytically dead Cas9$^{D10A/H840A}$, or dCas9, does not create DNA lesions and thus serves as a negative control. dCas9, however, can interfere with transcription when targeted to promoters and promoter-proximal coding region [50, 51]. Relying on this transcription-interfering property, a previous report from our laboratory established and characterized a dCas9-based CRISPR interference (CRISPRi) platform in *B. burgdorferi* [52]. Building on that work, we report herein the effects of targeting Cas9$^{WT}$ and its nickase versions to several *B. burgdorferi* endogenous plasmid loci.

## Materials and methods

### *E. coli* strains and growth conditions

*E. coli* host strain NEB 5-alpha F' *l$^q$* (New England Biolabs) was exclusively used to generate, store, and amplify the *E. coli*/*B. burgdorferi* shuttle vectors listed in Table 1. The resulting strains were grown on LB agar plates or in Super Broth (35 g/L bacto-tryptone, 20 g/L yeast extract, 5 g/L NaCl, and 6 mM NaOH) liquid medium with shaking at 30˚C [53]. Transformation was achieved by heat shock followed by recovery in SOC medium (New England Biolabs) for 1h at 30˚C with shaking. Antibiotic selection was achieved using spectinomycin at 50 μg/mL or rifampin at 25 μg/mL in liquid culture or 50 μg/mL in plates.

### *B. burgdorferi* strains and growth conditions

Previously described *B. burgdorferi* strain B31-A3-68-Δ*bbe02*::P$_{flgB}$-*aphI*, also known as K2, is an infectious, highly transformable derivative of the type strain B31 [54]. To derive strain CJW_Bb471 from K2, pseudogene *bbf29* of plasmid lp28-1 was disrupted by insertion of a gentamicin resistance cassette. Strains K2 and CJW_Bb471 contain 18 of the 21 endogenous plasmids of parental strain B31; they both lack endogenous plasmids cp9, lp5, and lp56 [6, 54]. To generate strain CJW_Bb471, 75 μg of plasmid p28-1::flgBp-aacC1 [34] were digested with *Age*I-HF (New England Biolabs), ethanol precipitated [55], resuspended in 25 μL water, and electroporated into a 100 μL aliquot of K2 electrocompetent cells. Electroporated cells were immediately transferred to 6 mL complete Barbour-Stoenner-Kelly (BSK)-II medium and allowed to recover overnight. The following day, cells were plated in semisolid BSK-agarose medium under kanamycin and gentamicin selection. A clone was grown and confirmed to have correct insertion of the gentamicin resistance cassette into lp28-1 and to contain all the endogenous plasmids of the parental strain.

*B. burgdorferi* strains were grown in complete BSK-II medium at 34˚C in a humidified 5% $CO_2$ incubator [56–58]. BSK-II medium contained 50 g/L bovine serum albumin (BSA), Universal Grade (Millipore), 9.7 g/L CMRL-1066 (US Biological), 5 g/L Neopeptone (Difco), 2 g/L Yeastolate (Difco), 6 g/L HEPES (Millipore), 5 g/L glucose (Sigma-Aldrich), 2.2 g/L sodium bicarbonate (Sigma-Aldrich), 0.8 g/L sodium pyruvate (Sigma-Aldrich), 0.7 g/L sodium citrate (Fisher Scientific), 0.4 g/L *N*-acetylglucosamine (Sigma-Aldrich), 60 mL/L heat-inactivated rabbit serum (Gibco), and had a pH of 7.6. For plating in semisolid BSK-agarose medium [52], each 10-cm plate was seeded with up to 1 mL *B. burgdorferi* culture. BSK-agarose plating medium was made by mixing two volumes of 1.7% agarose in water, melted and pre-equilibrated at 55˚C with three volumes of BSK-1.5 medium, also briefly (for less than 5 min) pre-equilibrated at 55˚C and containing appropriate amounts of antibiotics. Then, 25 mL of the BSK-agarose mix was added to each seeded plate, which was then gently swirled and allowed to solidify for ~30 min at room temperature in a biosafety cabinet. The plates were then transferred to a humidified 5% $CO_2$ incubator kept at 34˚C. BSK-1.5 medium contained 69.4 g/L

**Table 1.** *E. coli/B. burgdorferi* shuttle vectors[a] used in this study.

| Shuttle vector name | CJW strain number[b] | Selection[c] | Source or Reference |
|---|---|---|---|
| i. Shuttle vectors expressing catalytically inactive dCas9 | | | |
| pBbdCas9S | | Sm/Sp | [52] |
| pBbdCas9S_arr2 | | Sm/Sp, Rf | [52] |
| pBbdCas9S_P$_{syn}$-sgRNA500 | | Sm/Sp | [52] |
| pBbdCas9G_arr2 | | Gm, Rf | [52] |
| pBbdCas9S(RBSmut) | | Sm/Sp | [52] |
| pBbdCas9S(RBSmut)_arr2 | | Sm/Sp, Rf | [52] |
| pBbdCas9S(RBSmut)_ P$_{syn}$-sgRNA500 | | Sm/Sp | [52] |
| pBbdCas9S(RBSmut)_P$_{syn}$-sgRNAvlsE1 | CJW7267 | Sm/Sp | This study |
| pBbdCas9S(RBSmut)_P$_{syn}$-sgRNAvlsE2 | CJW7268 | Sm/Sp | This study |
| pBbdCas9S(RBSmut)_P$_{syn}$-sgRNAvls11 | CJW7269 | Sm/Sp | This study |
| pBbdCas9S(RBSmut)_P$_{syn}$-sgRNAbbf03 | CJW7282 | Sm/Sp | This study |
| pBbdCas9S(RBSmut)_P$_{syn}$-sgRNAbbe10 | CJW7280 | Sm/Sp | This study |
| pBbdCas9S(RBSmut)_P$_{syn}$-sgRNAbbe17 | CJW7281 | Sm/Sp | This study |
| ii. Shuttle vectors expressing the nickase Cas9$^{D10A}$ | | | |
| pBbCas9$^{D10A}$S(RBSmut) | CJW7290 | Sm/Sp | This study |
| pBbCas9$^{D10A}$S(RBSmut)_arr2 | CJW7291 | Sm/Sp, Rf | This study |
| pBbCas9$^{D10A}$S(RBSmut)_P$_{syn}$-sgRNA500 | CJW7292 | Sm/Sp | This study |
| pBbCas9$^{D10A}$S(RBSmut)_P$_{syn}$-sgRNAvlsE1 | CJW7293 | Sm/Sp | This study |
| pBbCas9$^{D10A}$S(RBSmut)_P$_{syn}$-sgRNAvlsE2 | CJW7294 | Sm/Sp | This study |
| pBbCas9$^{D10A}$S(RBSmut)_P$_{syn}$-sgRNAvls11 | CJW7295 | Sm/Sp | This study |
| pBbCas9$^{D10A}$S(RBSmut)_P$_{syn}$-sgRNAbbf03 | CJW7298 | Sm/Sp | This study |
| pBbCas9$^{D10A}$S(RBSmut)_P$_{syn}$-sgRNAbbe10 | CJW7296 | Sm/Sp | This study |
| pBbCas9$^{D10A}$S(RBSmut)_P$_{syn}$-sgRNAbbe17 | CJW7297 | Sm/Sp | This study |
| iii. Shuttle vectors expressing the nickase Cas9$^{H840A}$ | | | |
| pBbCas9$^{H840A}$S | CJW7108 | Sm/Sp | This study |
| pBbCas9$^{H840A}$S_arr2 | CJW7109 | Sm/Sp, Rf | This study |
| pBbCas9$^{H840A}$S_P$_{syn}$-sgRNA500 | CJW7110 | Sm/Sp | This study |
| pBbCas9$^{H840A}$S_P$_{syn}$-sgRNAvlsE1 | CJW7128 | Sm/Sp | This study |
| pBbCas9$^{H840A}$S_P$_{syn}$-sgRNAvlsE2 | CJW7129 | Sm/Sp | This study |
| pBbCas9$^{H840A}$S_P$_{syn}$-sgRNAvls11 | CJW7246 | Sm/Sp | This study |
| pBbCas9$^{H840A}$S_P$_{syn}$-sgRNAbbf03 | CJW7249 | Sm/Sp | This study |
| pBbCas9$^{H840A}$S_P$_{syn}$-sgRNAbbe10 | CJW7247 | Sm/Sp | This study |
| pBbCas9$^{H840A}$S_P$_{syn}$-sgRNAbbe17 | CJW7248 | Sm/Sp | This study |
| pBbCas9$^{H840A}$S(RBSmut) | CJW7155 | Sm/Sp | This study |
| pBbCas9$^{H840A}$S(RBSmut)_arr2 | CJW7156 | Sm/Sp, Rf | This study |
| pBbCas9$^{H840A}$S(RBSmut)_P$_{syn}$-sgRNA500 | CJW7157 | Sm/Sp | This study |
| pBbCas9$^{H840A}$S(RBSmut)_P$_{syn}$-sgRNAvlsE1 | CJW7158 | Sm/Sp | This study |
| pBbCas9$^{H840A}$S(RBSmut)_P$_{syn}$-sgRNAvlsE2 | CJW7159 | Sm/Sp | This study |
| pBbCas9$^{H840A}$S(RBSmut)_P$_{syn}$-sgRNAvls11 | CJW7250 | Sm/Sp | This study |
| pBbCas9$^{H840A}$S(RBSmut)_P$_{syn}$-sgRNAbbf03 | CJW7253 | Sm/Sp | This study |
| pBbCas9$^{H840A}$S(RBSmut)_P$_{syn}$-sgRNAbbe10 | CJW7251 | Sm/Sp | This study |
| pBbCas9$^{H840A}$S(RBSmut)_P$_{syn}$-sgRNAbbe17 | CJW7252 | Sm/Sp | This study |
| pBbCas9$^{H840A}$S(-10TC) | CJW7160 | Sm/Sp | This study |
| pBbCas9$^{H840A}$S(-10TC)_arr2 | CJW7161 | Sm/Sp, Rf | This study |
| pBbCas9$^{H840A}$S(-10TC)_P$_{syn}$-sgRNA500 | CJW7162 | Sm/Sp | This study |
| pBbCas9$^{H840A}$S(-10TC)_P$_{syn}$-sgRNAvlsE1 | CJW7163 | Sm/Sp | This study |

(*Continued*)

**Table 1.** (Continued)

| Shuttle vector name | CJW strain number[b] | Selection[c] | Source or Reference |
|---|---|---|---|
| pBbCas9$^{H840A}$S(-10TC)_P$_{syn}$-sgRNAvlsE2 | CJW7164 | Sm/Sp | This study |
| pBbCas9$^{H840A}$S(-10TC)_P$_{syn}$-sgRNAvls11 | CJW7254 | Sm/Sp | This study |
| pBbCas9$^{H840A}$S(-10TC)_P$_{syn}$-sgRNAbbe17 | CJW7255 | Sm/Sp | This study |
| iv. Shuttle vectors expressing wild-type Cas9 | | | |
| pBbCas9S(RBSmut) | CJW7283 | Sm/Sp | This study |
| pBbCas9S(RBSmut)_arr2 | CJW7284 | Sm/Sp, Rf | This study |
| pBbCas9S(RBSmut)_P$_{syn}$-sgRNA500 | CJW7285 | Sm/Sp | This study |
| pBbCas9S(RBSmut)_P$_{syn}$-sgRNAvlsE1 | CJW7286 | Sm/Sp | This study |
| pBbCas9S(RBSmut)_P$_{syn}$-sgRNAvls11 | CJW7278 | Sm/Sp | This study |
| pBbCas9S(RBSmut)_P$_{syn}$-sgRNAbbf03 | CJW7279 | Sm/Sp | This study |
| pBbCas9S(RBSmut)_P$_{syn}$-sgRNAbbe10 | CJW7288 | Sm/Sp | This study |
| pBbCas9S(RBSmut)_P$_{syn}$-sgRNAbbe17 | CJW7289 | Sm/Sp | This study |

[a]Naming of the *E. coli*/*B. burgdorferi* shuttle vectors follows the nomenclature established and described in detail in [52]. Of note, Cas9 variant expression is driven either by the IPTG-inducible P$_{pQE30}$ promoter or by its mutant versions in which the -10 region of the promoter (-10TC) or the ribosome binding site (RBSmut) were mutated to reduce basal Cas9 expression

[b]When requesting a plasmid from the Jacobs-Wagner lab, please include the CJW strain number alongside the plasmid name. For constructs previously published in [52], a CJW strain number is not provided, as the plasmids are available from Addgene. Please refer to the original publication for the Addgene catalog numbers

[c]Sm/Sp, streptomycin/spectinomycin resistance conferred by the *aadA* gene; Rf, rifampin resistance conferred by the *arr2* gene; Gm, gentamicin resistance conferred by the *aacC1* gene.

BSA, 12.7 g/L CMRL-1066, 6.9 g/L Neopeptone, 3.5 g/L Yeastolate, 8.3 g/L HEPES, 6.9 g/L glucose, 6.4 g/L sodium bicarbonate, 1.1 g/L sodium pyruvate, 1.0 g/L sodium citrate, 0.6 g/L *N*-acetylglucosamine, 40 mL/L heat-inactivated rabbit serum, and had a pH of 7.5. Antibiotics were used at the following concentrations: streptomycin at 100 µg/mL, gentamicin at 40 µg/mL, and kanamycin at 200 µg/mL [59–61]. Unless otherwise indicated, *B. burgdorferi* cultures were maintained in exponential growth by diluting cultures into fresh medium before cultures densities reached ~5 x 10$^7$ cells/mL. Cell density of cultures was determined by direct counting under darkfield illumination using disposable hemocytometers, as previously described [53].

## *B. burgdorferi* transformation, clone isolation, and characterization

Electrocompetent cells were generated as previously described [62] and stored as single use 50 or 100 µL aliquots at -80˚C. For shuttle vector transformations, 30 or 50 µg of plasmid eluted in water were electroporated (2.5 kV, 25 µF, 200 Ω, 2 mm gap cuvette) into 50 µL aliquots of competent cells. Electroporated cells were immediately transferred to 6 mL BSK-II and allowed to recover overnight. The next day, 100, 300, and 900 µL aliquots of the culture were each plated in semisolid BSK-agarose under selection. The remaining culture was diluted 6-fold in BSK-II and selected in liquid culture with appropriate antibiotics. Once transformants were observed as motile spirochetes, the liquid cultures were plated for clone isolation. Agarose plugs containing individual colonies were used to inoculate 6 mL BSK-II cultures. After 3 days, 500 to 1000 µL of each clonal culture was removed and pelleted at 10,000 x g for 10 min, the cells were resuspended and lysed in 50–100 µL water, and the resulting solution was used to perform multiplex PCR using primer pairs specific for each endogenous plasmid of strain B31 [63] and the DreamTaq Green DNA Polymerase (Thermo Scientific). For genomic DNA extraction, ~14 mL cultures were grown to ~10$^8$ cells/mL and then pelleted at 4,300 x g for 10 min at room temperature in a Beckman Coulter X-14R centrifuge equipped with a

swinging bucket rotor. The media was removed and the pellet was processed for DNA extraction using QIAGEN's DNeasy Blood & Tissue Kit protocol for Gram-negative bacteria. Final elution was carried out in 10 mM Tris-HCl, 0.1 mM EDTA, pH 9.0.

### Generation of *E. coli/B. burgdorferi* shuttle vectors for Cas9 and sgRNA expression

Table 1 lists the *E. coli/B. burgdorferi* shuttle vectors used or generated in this study. They were based on the previously described *B. burgdorferi* CRISPR interference platform [52]. The shuttle vectors express one of the following Cas9 versions: wild-type Cas9, the nickases Cas9$^{D10A}$ or Cas9$^{H840A}$, or the catalytically inactive dCas9 that carries both the D10A and H840A mutations. To revert the D10A mutation, site-directed mutagenesis was performed on appropriate template plasmids using Agilent's Quick Change Lightning Site-Directed Mutagenesis kit and primers NT651 and NT652. To revert the H840A mutation, site-directed mutagenesis was performed on appropriate template plasmids using primers NT749 and NT750. To generate plasmids with decreased basal expression of Cas9 proteins [52], site-directed mutagenesis was performed on appropriate plasmid templates using primers NT669 and NT670, which generated a weakened ribosomal binding site ("RBSmut" constructs), or primers NT677 and NT678, which introduced a mutation in the -10 region of the Cas9 promoter ("-10TC" constructs). Expression cassettes for the sgRNAs were moved among plasmids using restriction endonucleases *Asc*I and *Eag*I. To generate sgRNA expression cassettes, *Sap*I-digested Psyn-sgRNA500-containing plasmids were ligated with annealed primer pairs, as follows: primers NT657 and NT658 generated sgRNAvlsE1; NT660 and NT661 generated sgRNAvlsE2; NT721 and NT722 generated sgRNAvls11; NT723 and NT724 generated sgRNAbbe10; NT725 and NT726 generated sgRNAbbe17; and NT727 and NT728 generated sgRNAbbf03. Primer annealing was achieved by mixing 10 μL volumes of each primer at 5 μM concentration, then cycling the mix five times between 30 s at 95°C and 30 s at 55°C, followed by cooling to room temperature. Nucleotide sequences of primers used to generate the *E. coli/B. burgdorferi* shuttle vectors in this study are given in Table 2.

### DNA sequence analysis

To determine the sequence of the *vls* locus of *B. burgdorferi* strain K2, the 10910 base pair region encompassing *vlsE* and silent cassettes *vls2-vls16* was amplified using Platinum™ SuperFi™ DNA Polymerase (Thermo Fisher Scientific) and primers YN-LI_266 and YN-LI_267 (Table 2) and then sequenced with a SMRT Cell™ using 10-h data collection (Pacific Biosciences). The resulting reads were subjected to read-of-insert (ROI) analysis using SMRT Link v6.0.0 (Pacific Biosciences), followed by multiple sequence alignment using Geneious Prime 2019.0.4 (https://www.geneious.com), to obtain the final consensus sequence, which was deposited at GenBank under accession number OP620651.

### Data and material availability

*B. burgdorferi* strains and *E. coli/B. burgdorferi* shuttle vectors generated in this study (Table 1) are available upon request from Christine Jacobs-Wagner.

## Results

### Expression and targeting of Cas9 activity in *B. burgdorferi*

The previous report establishing CRISPRi in *B. burgdorferi* relied in part on all-in-one *E. coli / B. burgdorferi* shuttle vectors that carry a constitutive sgRNA expression cassette as well as an

**Table 2. Oligonucleotide primers used in this study.**

| Name | Sequence (5' to 3') |
| --- | --- |
| NT651 | GGATAAGAAATACTCAATAGGCTTAGATATCGGCACAAATAGCGTCGGATGGG |
| NT652 | CCCATCCGACGCTATTTGTGCCGATATCTAAGCCTATTGAGTATTTCTTATCC |
| NT657 | AGTGCTACAGGGGAGAATAATAA |
| NT658 | AACTTATTATTCTCCCCTGTAGC |
| NT660 | AGTGGATGGAGAGAAGCCTGAGG |
| NT661 | AACCCTCAGGCTTCTCTCCATCC |
| NT669 | GATAACAATTTCACACAGAATTCATTAAAGAAGAGAAATTACATATGGATAAGAAATAC |
| NT670 | GTATTTCTTATCCATATGTAATTTCTCTTCTTTAATGAATTCTGTGTGAAATTGTTATC |
| NT677 | GCTTTGTGAGCGGATAACAATTATAACAGATTCAATTGTGAGCGGATAACAATTTCACAC |
| NT678 | GTGTGAAATTGTTATCCGCTCACAATTGAATCTGTTATAATTGTTATCCGCTCACAAAGC |
| NT721 | AGTGCTGTTAGTGCTGGTTAGTG |
| NT722 | AACCACTAACCAGCACTAACAGC |
| NT723 | AGTAGGGGAAGACAATTTACTT |
| NT724 | AACAAGTAAATTGTCTTCCCCCT |
| NT725 | AGTAATATTCTTTCAGGGTAAGC |
| NT726 | AACGCTTACCCTGAAAGAATATT |
| NT727 | AGTAGAGTTTCTACGATTGAGTA |
| NT728 | AACTACTCAATCGTAGAAACTCT |
| NT749 | TAATCGTTTAAGTGATTATGATGTCGATCATATTGTTCCACAAAGTTTCCTTAAAGACG |
| NT750 | CGTCTTTAAGGAAACTTTGTGGAACAATATGATCGACATCATAATCACTTAAACGATTA |
| YN-LI_266 | GTATTTGTTGTTAAGTAGATAGGAATATTTCGG |
| YN-LI_267 | CGTGTCCATACACTTAATTAAAATCACTTATTC |

isopropyl β-D-1-thioglactopyranoside (IPTG)-inducible dCas9 expression cassette [52]. Using these CRISPRi shuttle vectors or control vectors that lack the sgRNA as background, we generated vectors (Fig 1) that express either Cas9$^{WT}$, which cleaves both DNA strands, or nickases Cas9$^{H840A}$ and Cas9$^{D10A}$, which cleave only one DNA strand [47].

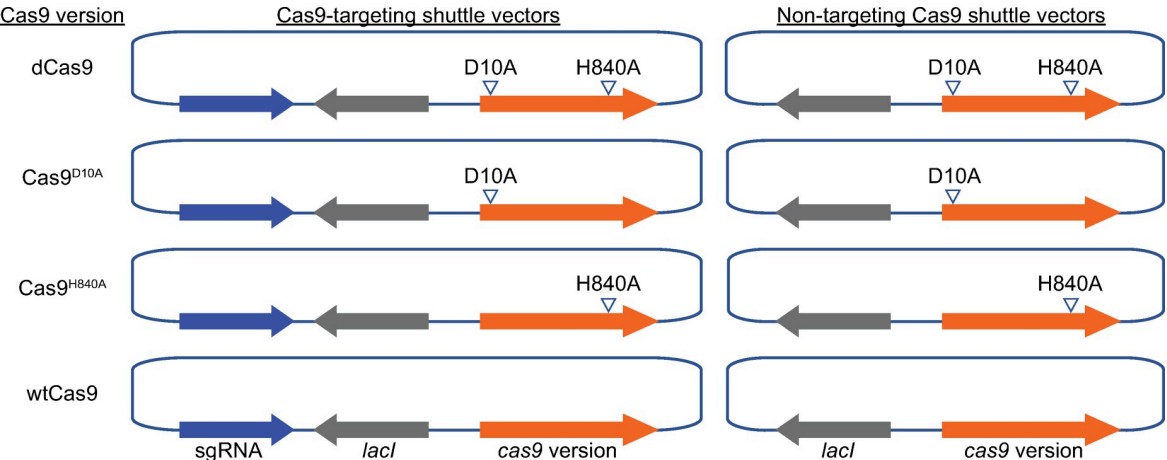

**Fig 1. Schematic depiction of *E. coli*/*B. burgdorferi* shuttle vectors used in this study.** Left, all-in-one, Cas9-targeting shuttle vectors carrying a sgRNA expression cassette as well as an IPTG-inducible Cas9 expression cassette that contains a constitutively expressed *lacI* gene. Right, non-targeting Cas9 shuttle vectors, which lack the sgRNA cassette. The Cas9 versions used are, from top to bottom: dCas9, Cas9$^{D10A}$, Cas9$^{H840A}$, and Cas9$^{WT}$. The dCas9 shuttle vectors were previously described [52]. Presence of the D10A or H840A mutation is indicated by arrowheads. Features are not drawn to scale. For simplicity, other important features of the shuttle vectors, such as the antibiotic resistance cassette or the *E. coli* or *B. burgdorferi* origins of replication, are not marked on the figure.

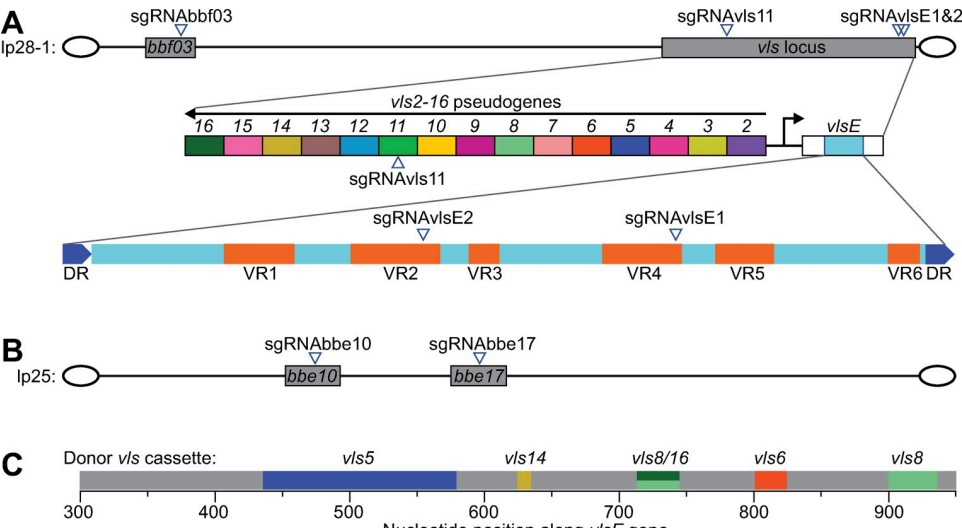

**Fig 2. Locations targeted by Cas9 activity in *B. burgdorferi* endogenous plasmids lp28-1 and lp25. A.** Top: schematic depiction of plasmid lp28-1. Marked (but not drawn to scale) are gene *bbf03* and the *vls* locus, which were targeted by the indicated sgRNAs. The sgRNAs were used one at a time, never in combination. Middle: magnification of the *vls* locus. Shown (but not drawn to scale) are the expressed *vlsE* lipoprotein gene and the 15 silent *vls* cassettes. Bottom: magnified view of the *vlsE1* cassette, which contains the variable regions of the expressed *vlsE* lipoprotein, flanked by two direct repeats (DRs). Variable regions (VRs) 1 through 6 are depicted, as well as the locations targeted by sgRNAvlsE1 and sgRNAvlsE2. Covalently closed hairpin telomeres are depicted as ovals flanking both ends of the linear plasmid. **B.** Same as in (A) but for plasmid lp25. Marked (but not drawn to scale) are genes *bbe10* and *bbe17*, which were independently targeted by the indicated sgRNAs. **C.** Depiction of part of the *vlsE* gene of strain K2. Shown in gray are sequences shared with the *vlsE* sequence reported for the parental strain B31. In color are divergent sequences that likely arose by recombination of the indicated silent cassettes into the expressed locus. The colors match those used for the silent cassettes in panel A. The *vls8/16* notation signifies a sequence that could have originated from either the *vls8* or *vls16* silent cassette.

In separate cultures, we targeted Cas9$^{WT}$ or its nickase versions to two endogenous *B. burgdorferi* plasmids, lp25 and lp28-1 (Fig 2). Plasmid lp25 encodes the nicotinamidase PncA which is essential for *B. burgdorferi*'s survival in the tick and vertebrate hosts [25, 34, 64–68]. Plasmid lp28-1 carries the *vls* antigenic variation system, which is composed of the expressed *vlsE* lipoprotein gene and 15 silent *vls* cassettes, *vls2-vls16* (Fig 2A), and is needed for the establishment of persistent infection in immunocompetent vertebrate hosts [11, 69, 70]. For lp28-1, we independently targeted two different sites in *vlsE*, plus one site in one of the silent *vls* cassettes, *vls11*, and another in the non-*vls* locus *bbf03* (Fig 2A). For lp25, we selected genes *bbe10* and *bbe17* and targeted them individually (Fig 2B). The sequences of the spacer and the PAM of these sgRNAs are listed in Table 3.

For these experiments, we used strain B31-A3-68 Δ*bbe02*::P*flgB*-*aphI*, also known as K2, a transformable, clonal, infectious derivative of the type strain B31 [54]. A mouse passage

**Table 3. sgRNAs used in this study.**

| sgRNA ID | Guide RNA spacer sequence (5' to 3') | PAM | *B. burgdorferi* target plasmid |
|---|---|---|---|
| bbe10 | AGGGGGAAGACAATTTACTT | TGG | lp25 |
| bbe17 | AATATTCTTTCAGGGTAAGC | AGG | lp25 |
| vlsE1 | GGATGGAGAGAAGCCTGAGG | AGG | lp28-1 |
| vlsE2 | GCTACAGGGGAGAATAATAA | AGG | lp28-1 |
| vls11 | GCTGTTAGTGCTGGTTAGTG | TGG | lp28-1 |
| bbf03 | AGAGTTTCTACGATTGAGTA | TGG | lp28-1 |

occurred during the derivation of strain K2 from the parental, sequenced B31 strain [31, 54]. During that mouse passage, gene conversion events likely changed the *vlsE* sequence. We therefore sequenced the entire *vls* locus of strain K2 using long read single-molecule, real-time (SMRT) sequencing [71] to obtain an accurate sequence encompassing the expressed *vlsE* gene and the repetitive silent *vls* cassettes. We found that the sequence of the silent *vls* cassette region was identical to the *B. burgdorferi* B31 reference sequence (GenBank accession number AE000794.2) [6]. In contrast, we found that the sequence of the *vlsE* gene of strain K2 had indeed diverged from the parental B31 *vlsE*, as expected. We detected five clusters of changes that could be attributed to segmental gene conversion events in which the original sequence was replaced by segments copied from the *vls2-vls16* silent cassette sequences (Fig 2C). Based on the alignment of the K2 *vlsE* sequence with the silent *vls2-16* cassette sequences, we designed two sgRNAs, sgRNAvlsE1 and sgRNAvlsE2, to maximize on-target (*vlsE*) and minimize off-target (the rest of the genome including *vls2-16*) binding potential (Fig 2C and Table 3). We were unable to generate a shuttle vector expressing both Cas9$^{WT}$ and sgRNAvlsE2, possibly due to toxicity of DSBs associated with off-target Cas9$^{WT}$ activity in *E. coli*. We did, however, generate shuttle vectors carrying genes encoding dCas9, Cas9$^{D10A}$, or Cas9$^{H840A}$, in combination with sgRNAvlsE2. The shuttle vectors containing these constructs are listed in Table 1.

## Targeting Cas9 activity to endogenous *B. burgdorferi* plasmids causes plasmid loss

We electroporated the shuttle vectors described above into strain K2. As controls, we used shuttle vectors lacking the sgRNA cassette and shuttle vectors expressing dCas9 rather than Cas9$^{WT}$ (Table 1). For each construct, we plated the electroporated cells after about three generations, grew a small number of the resulting clones, and determined their endogenous plasmid content by multiplex PCR, as previously described [63]. We found that all clones that had received a shuttle vector expressing Cas9$^{WT}$ and the *vlsE*-targeting sgRNAvlsE1 had lost the *vlsE*-carrying plasmid lp28-1 (Table 4, Fig 3). This was not due to widespread loss of lp28-1 from the parental strain, as clones obtained from electroporation of a shuttle vector expressing Cas9$^{WT}$ but no sgRNA retained their lp28-1 plasmid (Table 4, Fig 3). Similarly, electroporation of shuttle vectors encoding catalytically inactive dCas9, either alone or alongside sgRNAvlsE1 or sgRNAvlsE2, did not cause widespread lp28-1 loss (Table 4). The loss of lp28-1 occurred in spite of the presence of the adjacent homologous *vls2 –vls16* sequences that are used as donors for the generation of variant *vlsE* sequences during mammalian infection. There was also extensive plasmid loss when we targeted Cas9$^{WT}$ to two other sites on lp28-1: the silent cassette *vls11* or to the non-*vls* gene *bbf03* (Table 4, Fig 3). Therefore, Cas9$^{WT}$-mediated lp28-1 loss requires both Cas9 activity and targeting of this activity to the lp28-1 plasmid by a sgRNA regardless of where the DNA cut occurs. This effect was not limited to lp28-1, as targeting Cas9$^{WT}$ to genes *bbe10* or *bbe17* on endogenous plasmid lp25 resulted in loss of plasmid lp25 but not of lp28-1 (Table 4). All other endogenous *B. burgdorferi* plasmids were retained in almost all clones analyzed (Table 4). As with lp28-1, the loss of lp25 was dependent on Cas9 activity and the expression of a lp25-specific sgRNA, as expressing Cas9$^{WT}$ alone, or targeting dCas9 to lp25 did not affect lp25 retention (Table 4). We note that the transformants were selected and grown in the absence of Cas9 expression by IPTG induction. Presumably, the previously documented low but detectable basal expression of Cas9 from this system [52] generates enough activity to induce plasmid loss.

Performing multiplex PCR assays on individual clones is relatively labor-intensive. Additionally, if Cas9$^{WT}$-mediated plasmid loss is not 100% effective, the fraction of cells that still

**Table 4. Endogenous plasmid retention in individual *B. burgdorferi* clones after electroporation of Cas9/sgRNA vectors, as detected by multiplex PCR[a].**

| Cas9 version | Endogenous *B. burgdorferi* plasmid targeted | sgRNA | Clones analyzed | Plasmid retention (detected count / expected count for full retention)[b] | |
|---|---|---|---|---|---|
| | | | | Targeted plasmid (lp28-1 or lp25) | All other tested plasmids combined |
| Cas9[WT] | None | None | 4 | N/A[c] | 72/72 |
| | lp28-1 | vlsE1 | 4 | 0/4 | 68/68 |
| | | vls11 | 4 | 0/4 | 68/68 |
| | | bbf03 | 4 | 0/4 | 68/68 |
| | lp25 | bbe10 | 4 | 0/4 | 67/68 |
| | | bbe17 | 4 | 0/4 | 68/68 |
| dCas9 | None | None | 4 | N/A | 72/72 |
| | lp28-1 | vlsE1 | 4 | 4/4 | 68/68 |
| | | vlsE2 | 4 | 3/4 | 67/68 |
| | | vls11 | 4 | 4/4 | 68/68 |
| | | bbf03 | 4 | 4/4 | 68/68 |
| | lp25 | bbe10 | 4 | 4/4 | 68/68 |
| | | bbe17 | 4 | 4/4 | 68/68 |
| Cas9[D10A] | None | None | 4 | N/A | 70/72 |
| | lp28-1 | vlsE1 | 4 | 0/4 | 68/68 |
| | | vlsE2 | 4 | 0/4 | 68/68 |
| | | vls11 | 4 | 0/4 | 68/68 |
| | | bbf03 | 4 | 3/4 | 68/68 |
| | lp25 | bbe10 | 4 | 0/4 | 68/68 |
| | | bbe17 | 4 | 4/4 | 68/68 |
| Cas9[H840A] | None | None | 8 | N/A | 143/144 |
| | lp28-1 | vlsE1 | 16 | 0/16 | 272/272 |
| | | vlsE2 | 16 | 0/16 | 272/272 |
| | | vls11 | 4 | 0/4 | 68/68 |
| | | bbf03 | 4 | 2/4 | 68/68 |
| | lp25 | bbe10 | 4 | 0/4 | 68/68 |
| | | bbe17 | 4 | 4/4 | 68/68 |

[a]Data was aggregated based on the Cas9 version and the sgRNA expressed by the shuttle vector. Transformed strains carrying the same sgRNA but expressing different basal levels of the Cas9 variant were analyzed together. Plasmid detection was achieved by multiplex PCR [63]

[b]Data compares the number of endogenous plasmids detected in the analyzed clones with the expected number of endogenous plasmids if they had all been retained. All plasmid counts are combined for the non-targeted plasmids. A total of 18 non-targeted plasmids were assayed for each clone obtained by transformation with a shuttle vector lacking a sgRNA. A total of 17 non-targeted plasmids were assayed for each clone obtained by transformation with a shuttle vector expressing a sgRNA

[c]N/A, not applicable.

retain the targeted plasmid might be below detection. To avoid these drawbacks, we quantified endogenous plasmid retention by plating electroporated *B. burgdorferi* populations under differential antibiotic selection. In these plating assays, we used strain K2, in which retention of plasmid lp25 allows colony formation in the presence of kanamycin. Additionally, we derived strain CJW_Bb471 from strain K2 by inserting a gentamicin resistance cassette in its lp28-1 plasmid. This genetic modification does not interfere with *B. burgdorferi*'s ability to infect mice or be acquired by ticks [34]. Plating CJW_Bb471 transformants in the presence of kanamycin measures retention of lp25, while plating in the presence of gentamicin quantifies retention of lp28-1. In both cases, acquisition of streptomycin resistance indicates successful delivery of the Cas9-expressing shuttle vector. The number of streptomycin-resistant transformants detected in these experiments varied significantly both within an experiment and

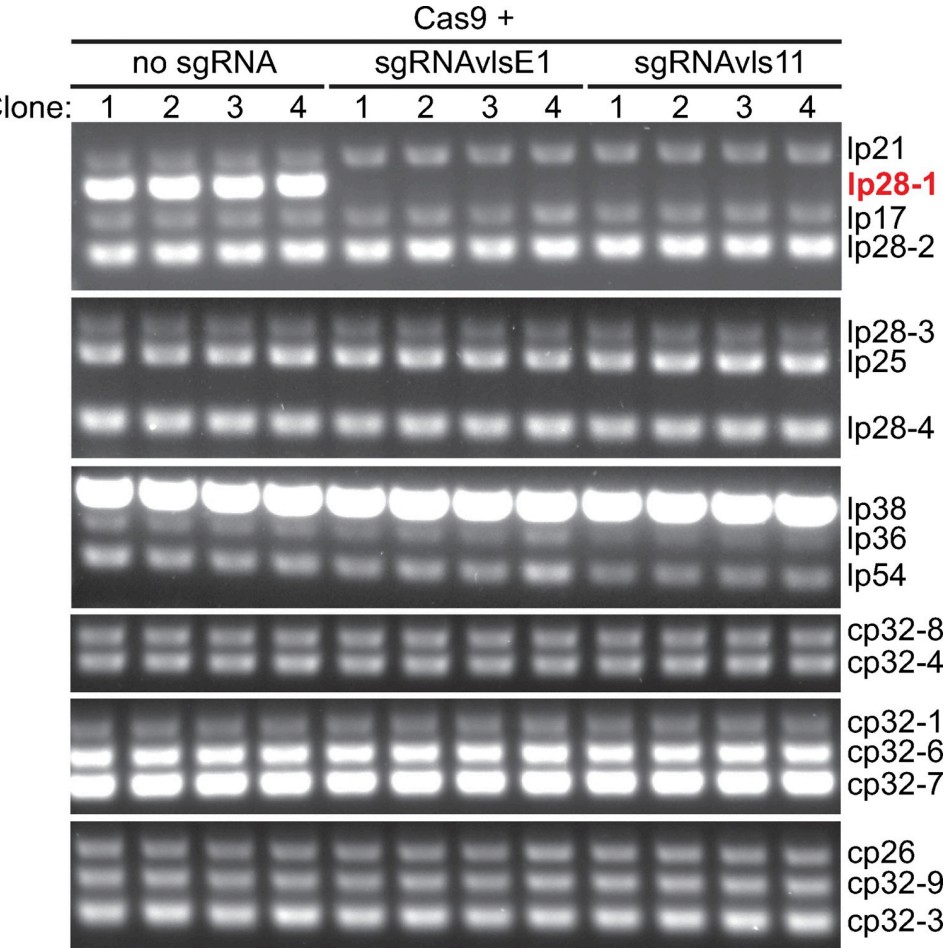

**Fig 3. Targeting Cas9 activity to lp28-1 causes the loss of this plasmid from the cell population.** Cells of *B. burgdorferi* strain K2 were electroporated with shuttle vectors expressing Cas9$^{WT}$ and the indicated sgRNAs. Four clones obtained from each transformation were analyzed by multiplex PCR for the presence or absence of *B. burgdorferi* endogenous plasmids. The PCR reactions were grouped into six sets. The endogenous plasmids corresponding to each of the bands are listed on the right. Signal intensity was scaled to ensure that all positive bands could be seen. As a result, the intensities of some bands are saturated. For uncropped source images of the gels, please see S1 Fig.

between experiments (S2A Fig), as did transformation frequencies, as measured in one of the experiments (S2B Fig). Despite these limitations, we could detect the following trends (see Tables 5 and 6 and Fig 4). First, targeting Cas9$^{WT}$ to either lp25 or lp28-1 did not result in the recovery of transformants retaining the targeted plasmid (Tables 5 and 6 and Fig 4). Second, targeting dCas9 did not cause plasmid loss (Tables 5 and 6 and Fig 4). Third, all Cas9 versions failed to induce loss of lp25 or lp28-1 in the absence of a targeting sgRNA (Tables 5 and 6 and Fig 4). Based on the number of clones that received the Cas9-expressing shuttle vector in each of the electroporations, we calculated the lowest frequency at which we could detect clones retaining the targeted endogenous plasmids (S2 Fig). These experimental limits of detection of plasmid retention within the transformed cell populations also varied significantly from electroporation to electroporation. The lowest limit of detection was around $10^{-3}$ (S2C Fig). This value indicates that DSB repair mechanisms that occur less frequently than in one cell out of 1,000 cells could not be detected in our assay.

**Table 5. Linear plasmid 25 (lp25) retention in *B. burgdorferi* following Cas9 targeting, as detected by plating.**

| Host strain (marked endogenous plasmid) | Cas9 version | sgRNA | Transformants detected by plating (CFU/mL)[a] | |
|---|---|---|---|---|
| | | | CFU/mL (selection for shuttle vector and endogenous plasmid) | CFU/mL (selection for shuttle vector alone) |
| K2 (lp25) Experiment 1 | Cas9$^{WT}$ | bbe10 | 0 | 910 |
| | | bbe17 | 0 | 500 |
| | dCas9 | bbe10 | 610 | 760 |
| | | bbe17 | 530 | 450 |
| | Cas9$^{D10A}$ | bbe10 | 118 | 810 |
| | | bbe17 | 370 | 530 |
| | Cas9$^{H840A}$ | bbe10 | 310 | 770 |
| | | bbe17 | 520 | 430 |
| CJW_Bb471 (lp25) Experiment 3[b] | Cas9$^{WT}$ | None | 10 | 6.9 |
| | | bbe10 | 0 | 12.3 |
| | | bbe17 | 0 | 17.7 |
| | dCas9 | None | 22.3 | 37.7 |
| | | bbe10 | 145 | 127.5 |
| | | bbe17 | 102.5 | 67.5 |
| | Cas9$^{D10A}$ | None | 82.5 | 107.5 |
| | | bbe17 | 202.5 | 160 |
| | Cas9$^{H840A}$ | None | 47.7 | 34.6 |
| | | bbe10 | 0 | 1.5 |
| | | bbe17 | 1360 | 1380 |

[a]Different volumes of transformant cultures were plated under streptomycin selection (which selects for the shuttle vector), or streptomycin + kanamycin selection (which selects for lp25). Colonies were counted after 2–3 weeks and the resulting count was used to calculate the concentration of selectable cells in the parental population of transformants, expressed as colony forming units (CFU) per mL

[b]Retention of both lp25 and lp28-1 was assayed in experiment 3 following electroporation of the indicated constructs. For this reason, results from this experiment are presented in both Tables 5 and 6.

## Effects of Cas9 nickases on *B. burgdorferi* endogenous plasmids

While Cas9$^{WT}$ robustly and specifically induced plasmid loss when targeted to lp25 or lp28-1 (Tables 4–6, Fig 4), the nickases Cas9$^{D10A}$ and Cas9$^{H840A}$ exhibited more heterogeneous behaviors. When analyzing by multiplex PCR the clones isolated in the absence of selection for the targeted plasmid, we found that targeting the nickases to the *vls* region of lp28-1 or the *bbe10* locus of lp25 was more efficient at causing plasmid loss than targeting the nickases to the *bbf03* locus of lp28-1 or the *bbe17* locus of lp25 (Table 4). We noticed a similar trend when we selected the transformants for the targeted plasmid (Tables 5 and 6, Fig 4). These differences could be due to distinct targeting efficiencies by the sgRNAs or could reflect varied efficiencies in repairing SSBs induced at the sgRNA-targeted location.

## Discussion

We previously showed that targeting dCas9 to selected *B. burgdorferi* genes causes specific and efficient downregulation of gene expression, allowing for relatively easy and fast strain generation and phenotypic investigation [52]. In this study, we show that targeting Cas9$^{WT}$ or its nickase variants to plasmid-encoded loci results in plasmid loss, though to a varying degree (Tables 4–6 and Fig 4). In the case of Cas9$^{WT}$, plasmid loss was very efficient, indicating that repair of double-stranded DNA breaks generated in this manner occurs below the detection limit of our assay, i.e., less than one in $10^3$ cells retained the targeted endogenous plasmid,

**Table 6. Linear plasmid 28–1 (lp28-1) retention in *B. burgdorferi* following Cas9 targeting, as detected by plating.**

| Host strain (marked endogenous plasmid) | Cas9 version | sgRNA | Transformants detected by plating (CFU/mL)[a] | |
|---|---|---|---|---|
| | | | CFU/mL (selection for shuttle vector and endogenous plasmid) | CFU/mL (selection for shuttle vector alone) |
| CJW_Bb471 (lp28-1) Experiment 2 | Cas9$^{WT}$ | None | 6.9 | 16.2 |
| | | vlsE1 | 0 | 5.4 |
| | dCas9 | None | 2.3 | 4.6 |
| | | vlsE1 | 6.9 | 4.6 |
| | | vlsE2 | 3.1 | 9.2 |
| | Cas9$^{D10A}$ | None | 2.3 | 1.5 |
| | | vlsE1 | 1.5 | 5.4 |
| | | vlsE2 | 3.1 | 4.6 |
| | Cas9$^{H840A}$ | None | 0.8 | 3.8 |
| | | vlsE1 | 0 | 14.6 |
| | | vlsE2 | 0 | 5.4 |
| CJW_Bb471 (lp28-1) Experiment 3[b] | Cas9$^{WT}$ | None | 13.8 | 6.9 |
| | | vlsE1 | 0 | 380 |
| | | vls11 | 0 | 16.1 |
| | | bbf03 | 0 | 29.2 |
| | dCas9 | None | 46.9 | 37.7 |
| | | vlsE1 | 35.4 | 39.2 |
| | | vlsE2 | 7.7 | 9.2 |
| | | vls11 | 29.2 | 16.1 |
| | | bbf03 | 11.5 | 5.4 |
| | Cas9$^{D10A}$ | None | 95 | 107.5 |
| | | vlsE2 | 0 | 7.7 |
| | | vls11 | 0 | 0.8 |
| | | bbf03 | 3.1 | 2.3 |
| | Cas9$^{H840A}$ | None | 51.5 | 34.6 |
| | | vlsE1 | 64.6 | 700 |
| | | vlsE2 | 58.5 | 270 |
| | | vls11 | 11.5 | 35 |
| | | bbf03 | 2.3 | 0.8 |

[a]Different volumes of transformant cultures were plated under streptomycin selection (which selects for the shuttle vector), or streptomycin + gentamicin (which selects for lp28-1). Colonies were counted and the resulting count was used to calculate the concentration of selectable cells in the parental population of transformants, expressed as colony forming units (CFU) per mL

[b]Retention of both lp25 and lp28-1 was assayed in experiment 3 following electroporation of the indicated constructs. For this reason, results from this experiment are presented in both Tables 5 and 6.

based on the highest number of transformants recovered after Cas9 shuttle vector electroporation (S2A and S2C Fig). The nickases Cas9$^{D10A}$ and Cas9$^{H840A}$ also cause significant plasmid loss. Presumably, a considerable fraction of nicked plasmids undergo degradation before DNA repair factors can be recruited to the site of the SSBs. Alternatively, repair of DNA lesions may be less efficient in *B. burgdorferi* compared to other bacteria, as several DNA repair factors (e.g., *mutH*, *lexA*, *ruvC*, *sbcB*, *recFOR*, *recX*) are absent from the *B. burgdorferi* genome [6, 72]. When considering the limit of detection of our assay (S2C Fig), our results suggest that the efficiency of DSB repairs in *B. burgdorferi* is at least below $10^{-3}$ even when donor sequences are present as in the case of *vlsE* and *vls11*. Further work will be required to gain better insight into the mechanisms employed by *B. burgdorferi* to repair DNA lesions.

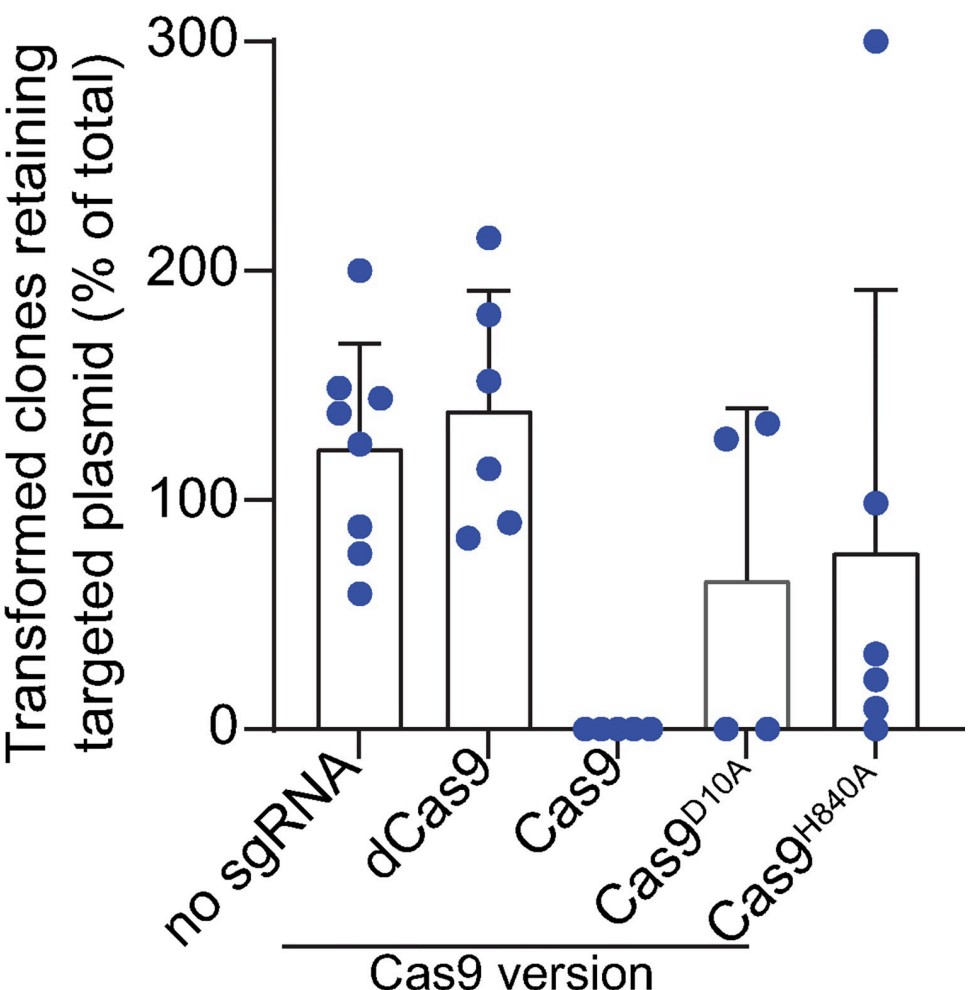

**Fig 4. Summary of the *B. burgdorferi* transformation results.** Graph compiling the plasmid retention values measured in experiment 3 described in Tables 5 and 6. Plasmid retention was calculated by dividing the concentration of cells that received the Cas9/sgRNA-expressing shuttle vector and retained the targeted plasmid by the concentration of cells that received the shuttle vector for any given electroporation. Experimental samples were grouped as follows. The "No sgRNA" group combines transformations of shuttle vectors encoding each of the four Cas9 versions (Cas9^WT, Cas9^D10A, Cas9^H840A, and dCas9) but no sgRNA. These transformations were plated under either kanamycin or gentamicin selection to assess retention of lp25 or lp28-1, respectively. All other transformations are grouped based on the version of Cas9 expressed from the shuttle vector and combine lp28-1 and lp25-targeting constructs.

Importantly, our work shows that targeting Cas9^WT to an endogenous *B. burgdorferi* plasmid is an easy and efficient method to displace the plasmid. The Cas9 nickases can also be used to achieve this outcome, but they are less effective. Our Cas9-based approach provides an alternative to the previously developed method that displaces endogenous plasmids through introduction of shuttle vectors belonging to the same plasmid compatibility class [33–39]. Both methods yield clones in which the targeted endogenous plasmid is replaced by a shuttle vector that carries an antibiotic resistance marker. The Cas9-based method, however, does not require prior knowledge of the targeted plasmid's replication and segregation locus [7, 33–39, 73, 74], and involves only an easy cloning step to insert the sgRNA sequence into the Cas9 shuttle vector. Additionally, as Cas9 activity can be simultaneously targeted to multiple locations in the genome by co-expression of relevant sgRNAs [51], simultaneous removal of multiple plasmids from a *B. burgdorferi* strain should be achievable via a single transformation.

While the degree of genome segmentation in Borreliaceae is the highest among the known bacteria, other bacteria have segmented genomes that can include circular and linear chromosomes, chromids, megaplasmids, as well as smaller plasmids [5]. Plasmids often encode virulence factors or antibiotic resistance genes and are stably maintained by highly effective plasmid segregation mechanisms that ensure faithful inheritance by daughter cells over generations [75]. The study of plasmid-encoded functions in bacteria other than the Lyme disease spirochetes can therefore be facilitated by implementation of a Cas9-mediated plasmid curation protocol. Translation of this approach across bacterial phyla is likely feasible, as demonstrated by the successful broad implementation of CRISPR-based methods of gene regulation [76].

## Supporting information

**S1 Fig. Original gel images used to generate Fig 3.**
(PDF)

**S2 Fig. Transformation statistics.**
(PDF)

## Acknowledgments

We thank Dr. Patricia Rosa for sharing strain K2 and plasmid p28-1::flgBp-aacC1, the members of the Jacobs-Wagner lab for critical reading of the manuscript, and Dr. George Chaconas for valuable discussions.

## Author Contributions

**Conceptualization:** Constantin N. Takacs, Yuko Nakajima, James E. Haber, Christine Jacobs-Wagner.

**Data curation:** Constantin N. Takacs, Yuko Nakajima.

**Formal analysis:** Constantin N. Takacs, Yuko Nakajima.

**Funding acquisition:** James E. Haber, Christine Jacobs-Wagner.

**Investigation:** Constantin N. Takacs, Yuko Nakajima.

**Methodology:** Yuko Nakajima.

**Project administration:** James E. Haber, Christine Jacobs-Wagner.

**Supervision:** James E. Haber, Christine Jacobs-Wagner.

**Validation:** Constantin N. Takacs, Yuko Nakajima.

**Visualization:** Constantin N. Takacs.

**Writing – original draft:** Constantin N. Takacs, Christine Jacobs-Wagner.

**Writing – review & editing:** Constantin N. Takacs, Yuko Nakajima, James E. Haber, Christine Jacobs-Wagner.

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
