## [Decision Letter · Decision Letter 0]

7 Oct 2022

PONE-D-22-25882Cas9-mediated endogenous plasmid loss in *Borrelia burgdorferi*PLOS ONE

Dear Dr. Jacobs-Wagner,

Thank you for submitting your manuscript to PLOS ONE. After careful consideration, we feel that it has merit but does not fully meet PLOS ONE’s publication criteria as it currently stands. Therefore, we invite you to submit a revised version of the manuscript that addresses each of the points raised during the review process.

We look forward to receiving your revised manuscript.

Kind regards,

Brian Stevenson, Ph.D.

Academic Editor

PLOS ONE

“We thank Dr. Patricia Rosa for sharing strain K2 and plasmid p28-1::flgBp-aacC1, the members of the Jacobs-Wagner lab for critical reading of the manuscript, and Dr. George Chaconas for valuable discussions. C.N.T. was supported in part by an American Heart Association postdoctoral fellowship (award number 18POST33990330). C.J.-W. is a Howard Hughes Medical Institute Investigator. Y.N. was supported by the Bay Area Lyme Foundation and the Brandeis University Provost’s Research Fund. J.E.H. was supported by NIH grant R35 GM127029. The funders had no role in study design, data collection, analysis, and interpretation, decision to submit the work for publication, or preparation of the manuscript.”

“C.N.T. was supported in part by an American Heart Association postdoctoral fellowship (award number 18POST33990330). C.J.-W. is a Howard Hughes Medical Institute Investigator. Y.N. was supported by the Bay Area Lyme Foundation and the Brandeis University Provost’s Research Fund. J.E.H. was supported by NIH grant R35 GM127029. The funders had no role in study design, data collection, analysis, and interpretation, decision to submit the work for publication, or preparation of the manuscript.”

Reviewers' comments:

Reviewer's Responses to Questions

**Comments to the Author**

1. Is the manuscript technically sound, and do the data support the conclusions?

Reviewer #1: Yes

Reviewer #2: Yes

2. Has the statistical analysis been performed appropriately and rigorously? 

Reviewer #1: Yes

Reviewer #2: Yes

3. Have the authors made all data underlying the findings in their manuscript fully available?

Reviewer #1: Yes

Reviewer #2: Yes

4. Is the manuscript presented in an intelligible fashion and written in standard English?

Reviewer #1: Yes

Reviewer #2: Yes

5. Review Comments to the Author

Reviewer #1: The manuscript by Takacs et al. describes a CRISPR-Cas9-based method to cure Borrelia burgdorferi (Bb) of some of its many plasmids. The authors build off their prior work using CRISPR-Cas to knock down Bb gene expression. Plasmids that are not required for growth in vitro (but are important in the natural tick-spirochete lifecycle) were successfully cured from Bb by wild type Cas9 nuclease that produces double strand breaks. Plasmids were also targeted by Cas9 point mutants (nickases) that produced single strand lesions on target plasmids.

The genetic tools described by the authors are a welcome addition to the field. The targeted elimination (by wt Cas9) or possible targeted suppression of plasmid copy number or destabilization of plasmids (by Cas9 nickases) will advance the field.

I only have a few minor comments and suggestions for the authors to consider.

1. line 33: Are humans accidental or incidental hosts?

2. Line 33: Isn’t Bb ‘always’ maintained (as opposed to ‘typically’) through a transmission cycle between vertebrates and ticks?

3. Line 64: “more targeted approach” seems more appropriate than “easier”

4. Line 68: Maybe I am misunderstanding, but isn’t a double strand break in the chromosome lethal to all (as opposed to several) Bacteria if not repaired?

5. Line 79: CRISPR-Cas is an adaptive bacterial immune system, not innate

6. Line 204-5: ‘Assays retention’ and ‘examines retention’ was confusing

7. Line 211 and elsewhere: Does plasmid destabilization mean something distinct from plasmid loss? If so, can the authors make this explicit?

8. Lines 210-: Please call out the figures showing data that supports claims in each of these sentences.

9. The methods are very detailed, thank you

10. Can the authors please provide the DNA sequence of the 10,910 base pair regions encompassing vlsE and silent cassettes vls2-vls16 in K2 and B31? These sequences may be useful to labs studying antigenic variation in Bb.

Reviewer #2: The authors of this study have developed a method to selectively target individual plasmids of the Lyme disease bacterium. The authors rely on an inducible cas9 based system to nick targeted DNA via sgRNAs, and show that the targeted plasmids have been lost, while retaining the other endogenous plasmids. They demonstrate that this is most efficient with wild-type cas9 that nicks both DNA strands, and less efficient when either of two mutant cas9s that only nick one strand of DNA are used.

This novel method of removing individual plasmids will be broadly useful for researchers studying B. burgdorferi, and the functions that specific plasmids have in culture as well as during the tick-mouse life cycle.

The data are sound, and the manuscript should be accepted as is.

6. PLOS authors have the option to publish the peer review history of their article (what does this mean?). If published, this will include your full peer review and any attached files.

Reviewer #1: No

Reviewer #2: No

---

## [Author Response · Author response to Decision Letter 0]

3 Nov 2022

We have uploaded a "Response to Reviewers" file containing our response to the reviewer's comments.

---

## [Editor Report · Decision Letter 1]

11 Nov 2022

Cas9-mediated endogenous plasmid loss in *Borrelia burgdorferi*

PONE-D-22-25882R1

Dear Dr. Jacobs-Wagner,

We’re pleased to inform you that your manuscript has been judged scientifically suitable for publication and will be formally accepted for publication once it meets all outstanding technical requirements.

Kind regards,

Brian Stevenson, Ph.D.

Academic Editor

PLOS ONE
---

## [Editor Report · Acceptance letter]

16 Nov 2022

PONE-D-22-25882R1 

Cas9-mediated endogenous plasmid loss in *Borrelia burgdorferi*

Dear Dr. Jacobs-Wagner:

I'm pleased to inform you that your manuscript has been deemed suitable for publication in PLOS ONE. Congratulations! Your manuscript is now with our production department. 

Kind regards, 

on behalf of

Prof. Brian Stevenson 

Academic Editor

PLOS ONE